# Novel near-infrared emission from crystal defects in MoS$_2$ multilayer flakes

F. Fabbri[1,2], E. Rotunno[1], E. Cinquanta[3], D. Campi[4], E. Bonnini[1], D. Kaplan[5], L. Lazzarini[1], M. Bernasconi[4], C. Ferrari[1], M. Longo[3], G. Nicotra[6], A. Molle[3], V. Swaminathan[5] & G. Salviati[1]

The structural defects in two-dimensional transition metal dichalcogenides, including point defects, dislocations and grain boundaries, are scarcely considered regarding their potential to manipulate the electrical and optical properties of this class of materials, notwithstanding the significant advances already made. Indeed, impurities and vacancies may influence the exciton population, create disorder-induced localization, as well as modify the electrical behaviour of the material. Here we report on the experimental evidence, confirmed by *ab initio* calculations, that sulfur vacancies give rise to a novel near-infrared emission peak around 0.75 eV in exfoliated MoS$_2$ flakes. In addition, we demonstrate an excess of sulfur vacancies at the flake's edges by means of cathodoluminescence mapping, aberration-corrected transmission electron microscopy imaging and electron energy loss analyses. Moreover, we show that ripplocations, extended line defects peculiar to this material, broaden and redshift the MoS$_2$ indirect bandgap emission.

[1] IMEM-CNR Institute, Parco Area delle Scienze 37/A, 43124 Parma, Italy. [2] KET Lab, c/o Italian Space Agency via del Politecnico, 00133 Roma, Italy. [3] Laboratorio MDM, IMM-CNR, via C. Olivetti 2, I-20864 Agrate Brianza, Italy. [4] Dipartimento di Scienza dei Materiali, Università di Milano-Bicocca, Via R. Cozzi 55, 20126 Milano, Italy. [5] U.S. Army RDECOM-ARDEC, Fuze Precision Armaments and Technology Directorate, Picatinny Arsenal, New Jersey 07806-5000, USA. [6] IMM-CNR Institute, Strada VIII, 5, 95121 Catania, Italy. Correspondence and requests for materials should be addressed to G.S. (email: giancarlo.salviati@cnr.it).

Semiconducting transition metal dichalcogenide (TMD) monolayers, like MoS$_2$ or WS$_2$, have been proposed as promising channel materials for field-effect transistors[1,2]. Their high mechanical flexibility, stability and quality coupled with potentially inexpensive production methods offer prospective advantages compared with organic and crystalline bulk semiconductors. One of the advantages of two-dimensional (2D) TMDs, for example, with respect to graphene, comes from quantum confinement, enabling the indirect-to-direct bandgap transition as a function of the thickness[3,4]. Actually, with decreasing thickness, the indirect bandgap (1.29 eV), which lies below the direct gap in the bulk material, blueshifts in energy. This leads to a crossover to a direct-gap material (1.8 eV) in the limit of the single monolayer[3–5]. This particular effect can lead to a strong interaction with light, which can pave the way for envisioning the next generation of visible light-emitting devices. The analysis of the extended crystal defects, either intrinsic or extrinsic, in 2D nanoflakes is still in its early stage and deserves a focused approach to understand how novel electronic and/or optical properties can be engineered by controlling the nucleation of extended defects. In this respect, most of the consideration was devoted to the generation and/or the structural properties of extended defects[6–8]. For instance, ripplocations[9], line defects with a dual nature of surface ripple and crystallographic dislocation, have been studied only from a structural point of view, while their optical and electrical properties currently remain unexplored. Only in the case of a few extended defect types were some peculiar and beneficial optical or electrical properties reported, including the recently observed dislocation-induced memristive behaviour in MoS$_2$, the edge-enhanced photoluminescence response in WS$_2$, and the conductivity bias induced by Se-poor mirror twin boundaries[10–12].

In this work we report on the experimental evidence of a near-infrared (NIR) emission from crystalline defects in MoS$_2$ multilayer flakes exfoliated from bulk molybdenite. Cathodoluminescence (CL) spectroscopy and mapping reveal that the MoS$_2$ flake's edges present an intense emission in the NIR range (peaked at about 0.75 eV). According to electron energy loss spectroscopy and mapping (EELS) results and *ab initio* calculations of the defect-related intra-bandgap states energies, the origin of this emission is ascribed to the high concentration of sulfur vacancies ($V_s$). High-resolution transmission electron microscopy (TEM) and Raman spectroscopy and mapping show the presence of ripplocations and assess the defective behaviour of the MoS$_2$ flake's edges. In particular, the ripplocations induce a strong redshift and broadening of the indirect band-to-band transition of MoS$_2$, peaked at 1.25 eV as evidenced by CL spectroscopy and mapping.

Despite the great attention devoted in the literature to MoS$_2$ monolayers, we believe that our results will convince the reader that MoS$_2$ multilayer flakes can be considered good candidates for future technological applications, provided tailored defect engineering is achieved. Indeed, since the chalcogen vacancy has the lowest formation energy for all the 2D TMDCs[13], our findings have a general validity making the field of MX$_2$ (M: metal; X: chalcogenide) flakes with thickness $t > 1$ monolayer fertile for future investigations and emerging technological applications with accurately tailored properties.

## Results

**CL spectroscopy of MoS$_2$ flakes.** Figure 1 shows the CL spectroscopy and mapping study of a typical single exfoliated multilayer MoS$_2$ flake. Figure 1a shows the flake's secondary electron image whose CL spectrum (red solid line), integrated over the whole flake, is reported in Fig. 1b. The flake spectrum presents a broad emission set at 1.07 eV and a sharp one at 0.75 eV (see Gaussian deconvolutions reported in Supplementary Fig. 1 and Supplementary Note 1). The pristine molybdenite spectrum (blue dashed line in Fig. 1b) is shown as a reference (Supplementary Fig. 2 and Supplementary Note 1). The apparent redshift and broadening of the MoS$_2$ band-to-band transition with respect to the pristine material is due to the presence of a novel emission at 0.98 eV. The peaks at 1.25 and 1.10 eV are consistent with the known pristine molybdenite optical emissions, while the 0.98 eV peak appears only after the mechanical exfoliation process and therefore we can suppose that it is related to the defects formed during this process. It is worth noticing that the integrated intensities of the 0.75 eV emission in the flakes spectrum and the 1.25 eV emission in the pristine MoS$_2$ spectrum are comparable. In particular, the 0.75 eV emission integrated intensity is 30% higher (Supplementary Fig. 3 and Supplementary Note 1). In addition, the stability of the light emissions under electron beam irradiation is tested (Supplementary Fig. 4 and Supplementary Note 1) by means of CL spectroscopy after a 30 min of irradiation of a MoS$_2$ flake.

The spectral resolution of the CL system used in this study is $\pm 50$ meV. The accurate Gaussian deconvolution procedures reported in Supplementary Note 1 are outside the error bar allowing to confidently distinguish between the peaks underneath the broad band centred at 1.07 eV. However, both the excitation nature of the CL (highly energetic electrons) and the indirect type of the transitions broaden the peaks full width at half maximum (FWHM) reducing the CL resolution due to the high number of phonons generated inside the crystal[14]. The CL spectra were deconvoluted using a standard Levenberg–Marquardt algorithm to minimize the $\chi^2$. As a result of the fitting, all peak positions are affected by an error of 0.01 eV, which is less than the error from the spectral resolution of the measurement (0.05 eV or 5 nm). The area and the FWHM have an approximate error of 5%. Considering the aforementioned points, the peak energy assignation is the most accurate possible in the experimental limit of the technique. Figure 1c,d shows the CL monochromatic maps obtained by selecting the 0.75 and 1.07 eV bands, respectively. There is a clear spatial anticorrelation between the maps: the 0.75 eV emission is mainly localized on the edges of the flake, meanwhile the 1.07 eV transition is widespread throughout the whole area of the flake. This is due to the presence of the ripplocations all across the flakes as shown by scanning transmission electron microscopy (STEM) imaging in Supplementary Fig. 5. For this reason it is not possible to spatially resolve the 0.98 eV peak because the minority carrier diffusion length of our flakes is larger than the average distance among the ripplocations. In addition, the CL map acquired at 1.07 eV also contains the spectral features due to the bulk molybdenite. Therefore, due to the superimposition of the two emissions at 0.98 and 1.10 eV (Supplementary Fig. 1 and Supplementary Note 1), it is not possible to show the spatial distribution of the different components in CL imaging mode with the experimental parameters used (see Methods, CL spectroscopy).

As a conclusion of this part of the discussion, it must be stressed that this observation is the first experimental proof of the influence of ripplocations on the optical emission of multilayer MoS$_2$ flakes. The absence in the literature of similar results could be probably ascribed to different defect concentrations in the studied flakes with respect to our case.

**Origins of the emission at 0.75 eV.** Figure 2a shows an atomic resolution STEM-high-angle annular dark-field (HAADF) image of the edge of the MoS$_2$ exfoliated flake to investigate the

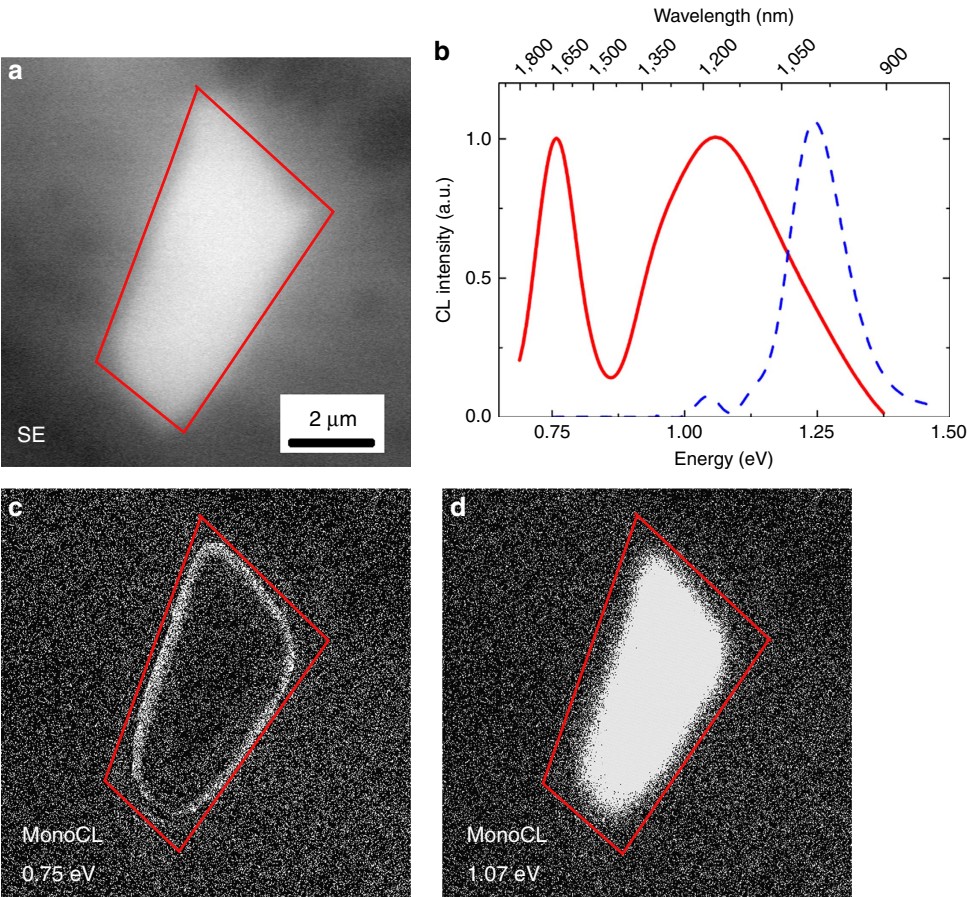

**Figure 1 | Scanning electron microscopy (SEM) secondary image and CL spectroscopy and monochromatic maps of a typical single exfoliated multilayer MoS₂ flake.** (**a**) Secondary electron SEM image of the MoS₂ micrometric flake. (**b**) CL spectrum of the MoS₂ flake (solid line) compared with the spectrum of molybdenite as a reference (dashed line). (**c,d**) Monochromatic maps at 0.75 and 1.07 eV, respectively.

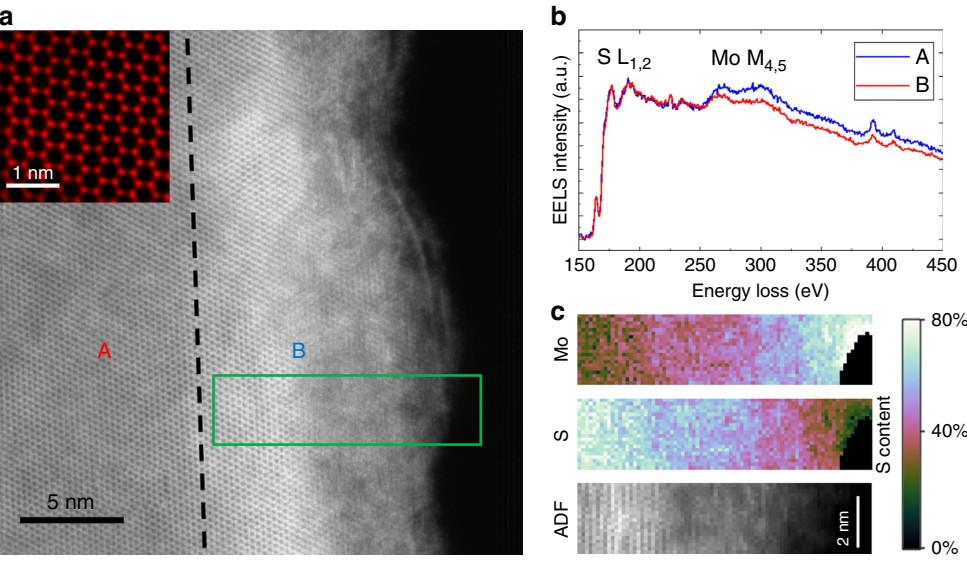

**Figure 2 | HAADF-STEM image and EELS spectroscopy and imaging of the edge of an exfoliated MoS₂ flake.** (**a**) HAADF-STEM image of the edge of an exfoliated MoS₂ flake and its atomically resolved structure reported in the inset. (**b**) EELS spectra obtained in the two positions marked A and B in **a** with the same number/colour code. (**c**) EELS spectrum images of the green rectangle in **a**.

possible presence of edge-related defects accounting for the observed optical emissions. We evaluated the sample thickness by analysing the different grey levels in STEM micrographs, corresponding to different number of monolayers. The maximum thickness measured for the studied flakes, that is, in the specimen centre, does not exceed 30 monolayers (18 nm) and typically

ranges between 30 and a few monolayers going towards the flakes borders. However, from Supplementary Fig. 6 and Supplementary Note 2, it can be clearly observed that the lateral size of the thickness decreasing region is about 200 nm. This dimension is comparable to the region where the 0.75 eV emission is recorded. Therefore, we can conclude that the infrared emission is a general feature of multilayer flakes, independent of the actual flake thickness.

The image is divided in two regions with different contrast by the black dashed line. On the left, region A, the sample is characterized by uniform contrast without any defects and a very well-defined atomic resolution pattern. The inset shows a higher magnification demonstrating the perfect hexagonal symmetry of the $MoS_2$ basal plane. On the right side, region B, we find a sudden enhancement of the image intensity followed by a blurred region where the atomic resolution is lost as the flake is no longer in a suitable zone axis. The augmented intensity comes from the increased thickness of the sample due to the wrapping of the foil on itself (Supplementary Figs 7 and 8) that usually occurs in the proximity of the edges (within 10–20 nm from the edge).

Figure 2b reports two EELS spectra where the S-$L_{1,2}$ and the Mo-$M_{4,5}$ edges are present. Both the spectra have been normalized to the S peak as the Mo edge is located on the tail of the S signal. The spectrum in red has been recorded in the defect-free area at the point labelled A in Fig. 2a. The quantitative analysis finds the ratio Mo/S = 0.51, very close to the exact $MoS_2$ stoichiometry (Supplementary Fig. 9). The second spectrum (blue line) comes from the wrapped region (point B). Here the Mo-$M_{4,5}$ edge is more intense with respect to the other spectrum. The quantitative analysis gives Mo/S = 0.55, confirming a significant S deficiency. Given this value, the only way to discriminate a S deficiency from an increase in the Mo content is to perform an absolute measurement. However, this kind of measurement requires to know precisely the sample thickness but, being the flake edge wrapped, its thickness is constantly changing making the absolute quantification unfeasible.

In any case, from a chemical point of view an increased Mo content is unlikely. In fact, a Mo increase can come only from external sources while no artefacts have been induced during the sample preparation (tape exfoliation) and TEM investigation (flakes deposited on a copper grid supported by a pure steel specimen holder with no Mo inside). At the same time the energy formation of the S vacancy is known to be the smallest compared with other intrinsic point defects in $MoS_2$ (ref. 6). These observations lead us to conclude that the variation in the Mo/S ratio is due only to the presence of S vacancies.

In the S and Mo spectrum imaging of Fig. 2c, recorded in the green rectangle in Fig. 2a, the gradual S depletion from the centre to the edge of the flake clearly appears.

Sulfur vacancies are known to have the lowest formation energy compared with other intrinsic point defects in $MoS_2$ (refs 6,15); therefore, they are most likely to be found at the flake edges, where energy is released during mechanical exfoliation. We can therefore deduce that the massive S deficiency can induce wrapping of the flake edges due to the formation of vacancy-related extended defects[15]. Similar results have been reported for different TMD systems, namely $MoSe_2$, where the presence of Se deficiency-induced mirror twin boundaries causes the wrapping of the flakes[12]. The calculations of charge transition level within density functional theory (DFT) with a hybrid exchange-correlation functional[16] in ref. 15 revealed that the single sulfur vacancy induces an acceptor state at about 0.77 eV below the conduction band for bulk $MoS_2$. This energy can be taken as an estimate of the adiabatic transition energy from the conduction band to the defect state, which, in the case of a small Franck–Condon shift, would be comparable to the vertical transition energy measured by CL. Indeed, the attribution of the 0.75 eV emission to sulfur vacancy-related intra-bandgap states is also supported by the spot-mode CL analysis, reported in Supplementary Fig. 10, and by the substoichiometric character of molybdenite, reported in Supplementary Fig. 9. This analysis confirms that the edge localization of the 0.75 eV emission is due to a high concentration of sulfur vacancies at the flake edges.

To investigate the dependence of the emission at 0.75 eV on the vacancy content, we computed the electronic band structures for two models of $MoS_2$ with concentrations of sulfur vacancies of 2.1 atom% (1/48) or 3.7 atom% (1/27). The models are built by inserting a sulfur vacancy in each $MoS_2$ lamella of a $4 \times 4$ or $3 \times 3$ supercell in the ab-plane of the hexagonal crystal. The generalized gradient corrected approximation of the exchange and correlation functional proposed by Perdew–Burke–Ernzerhof (PBE)[17] has been used here instead of the more computationally demanding hybrid functional used in ref. 15 (see Methods, DFT calculations). This choice leads to an underestimation of the bandgap and of the energy for the transition from the conduction band to the acceptor state due to vacancies as shown in Fig. 3. Still the calculation demonstrates that the blueshift of the emission at 0.75 eV can be due to an increased vacancy concentration.

The weakly dispersed bands at around 0.3 eV in Fig. 3 are empty states localized on sulfur vacancies. By increasing the vacancy content, the dispersion of these defect bands increases, leading to an increase in the separation between the bottom of the

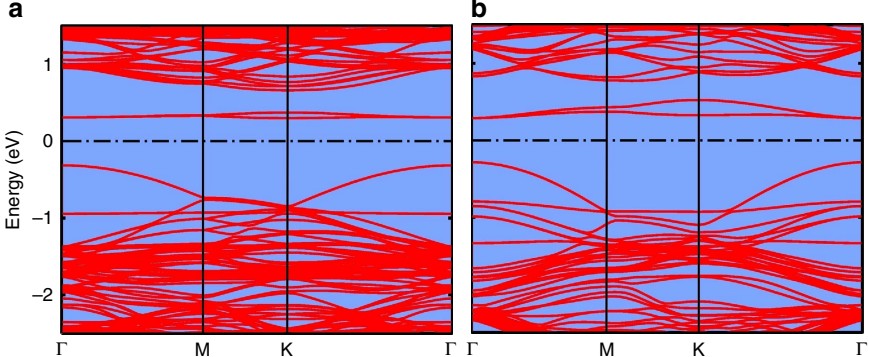

**Figure 3 | Electronic band structure of $MoS_2$ with sulfur vacancies.** Electronic band structures resulting from DFT calculations of models of $MoS_2$ with concentrations of sulfur vacancies of (**a**) 2.1 atom% (1/48) and (**b**) 3.7 atom% (1/27). The zero of the energy axis is the Fermi level, that is, at midgap between the occupied and empty states. Note that the Brillouin zone is different in the two models, and thus the scale of the horizontal axis is different in the two panels.

conduction bands (at around 0.7 eV in Fig. 3) and the centre of the defect bands. The disordered distribution of vacancies is expected to break the selection rule on the crystal momentum, and transitions from the bottom of the conduction band to the entire defects band are expected. By including the full width of the defects band, the DFT-PBE energy of this transition increases from the 0.28–0.35 eV range of the model with a vacancy content of 2.1 atom% to the 0.25–0.48 eV range in the model with 3.7 atom% of vacancies. Therefore, the DFT-PBE calculations forecast the blueshift of this emission with increasing vacancy content, although they do not exactly predict the bandgap width and the transition energy. The same effect is also present at much larger vacancy contents up to about 10 atom% as reported in ref. 18.

**Origins of the emission at 1.07 eV.** As far as the emission band set at 1.07 eV is concerned, the comparison between the light emission from the pristine molybdenite and $MoS_2$ multilayer flakes reveals that the indirect band-to-band transition turns out to be broadened and redshifted in the flakes. This observation can be related to the presence of a large number of ripplocations in the flakes (Fig. 4a), stemming from the mechanical exfoliation. They appear as straight, sharp and bright lines on the flake surface. It should be noted that not all the ripplocations show the same contrast, in fact, according to the existing literature[9], ripplocations having the same sign can attract each other and merge. In this way they reduce the overall energy while increasing their distortion field. Since the increasing number of extra planes close together in the dislocation core increases the local density it

can be assumed that the larger the dislocation core the higher the number of extra planes it contains.

The largest ripplocations are aligned to well-defined crystallographic orientations, perpendicular to the [10–10] directions, whilst other lower-order ripplocations can have different orientations. This is consistent with a previous report[9] that indicated the [10–10] (see Fig. 4c) direction as the preferential slip plane for ripplocations in $MoS_2$.

It is interesting to observe that the Raman spectra change as a function of the position of the laser spot on the flake. Figure 4d shows the comparison between the Raman spectra acquired in the flake centre, on the same ripplocation observed in the TEM image reported in Fig. 4b and at the flake edge. Intriguingly, the intensity ratio between the $A_{1g}$ and $E_{2g}$ modes varies according to the spot location. The $E_{2g}$ mode is more intense in the flake centre as commonly observed in geological flakes[19]. Remarkably, the Raman spectrum of the ripplocation shows an intensity ratio amounting to $\sim 1$ reflecting the presence of the defect, as far as the ripplocation-induced strain is expected to influence the in-plane $E_{2g}$ mode mostly.

A more intense modification of the $A_{1g}/E_{2g}$ intensity ratio occurs at the flake edges, where the ratio is inverted with respect to the flake centre. The $A_{1g}/E_{2g}$ intensity ratio map reported in Fig. 4e shows that this parameter can be considered as a fingerprint of the presence of crystal defects in mechanically exfoliated $MoS_2$ flakes. Indeed, following Parkin et al.[20], a change in the $A_{1g}/E_{2g}$ intensity ratio can be ascribed to the presence of sulfur vacancies. It is worth noticing that this observation is in excellent agreement with the CL monochromatic map reported in Fig. 1c.

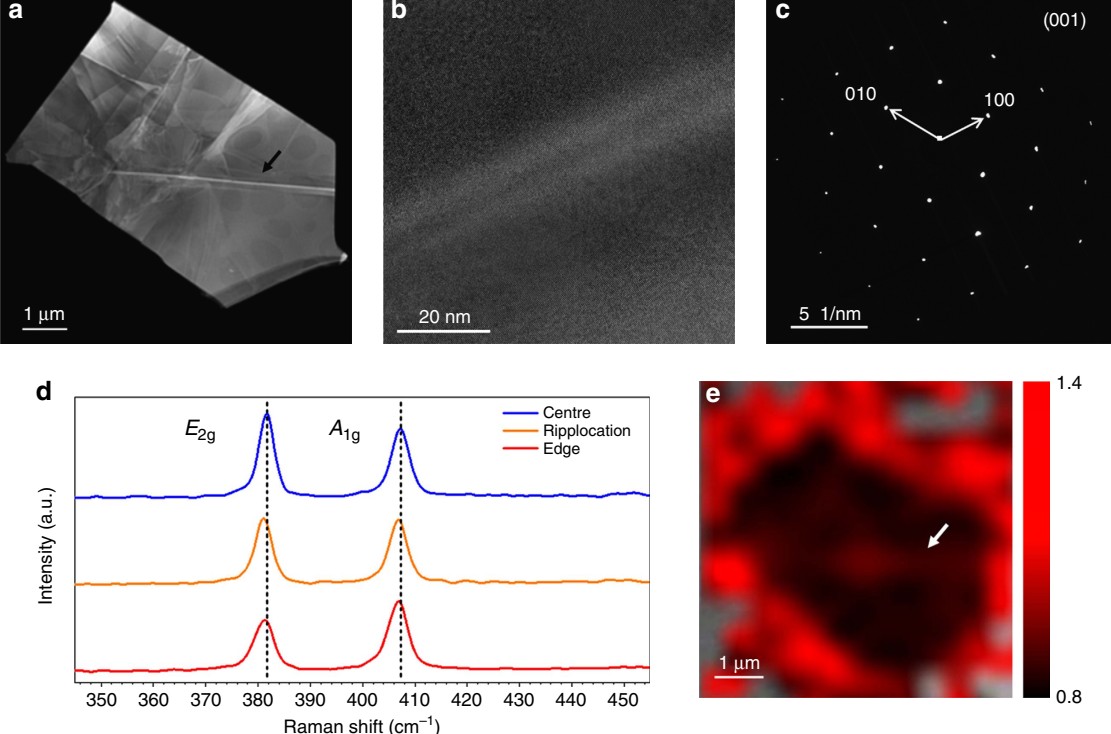

**Figure 4 | STEM imaging and Raman spectroscopy and imaging of ripplocations. (a)** STEM micrograph of the flake with an estimated size of $5 \times 3\,\mu m$. **(b)** High-resolution TEM image of the typical ripplocation. The atomic structure appears unaltered along the defect thus confirming the 'ripple' nature of the dislocation. **(c)** Diffraction Pattern of the flake. **(d)** Raman spectra of a bulk $MoS_2$ flake acquired in the flake centre (blue curve), on the ripplocation (orange curve) and at the flake edge (red curve). **(e)** $A_{1g}/E_{2g}$ peak intensity ratio map of the flake reported in **a**, where the region outside the flake are kept transparent (grey). Notice the different ratio between the edge and the central region of the flake. The black and white arrows have been added as guides to the eye in marking a ripplocation in the TEM image and Raman map, respectively.

The presence of the ripplocations can account for the onset of the emission set at 1.07 eV according to two main effects. Either the ripplocations induce a strain field that might affect the optical light emission properties of $MoS_2$ or the presence of ripplocations causes the formation of shallow states in the $MoS_2$ bandgap.

The Gaussian deconvolution analysis of the CL spectra of the asymmetric band peaked at 1.07 eV (Supplementary Note 1) supports the second hypothesis. This asymmetry is due to the presence of additional peaks at 1.25, 1.10 and 0.98 eV. The peaks at 1.25 and 1.1 eV correspond to the two components found in the pristine molybdenite spectrum. Therefore, we can ascribe the peak around 0.98 eV to intra-bandgap states induced by ripplocations formed during the mechanical exfoliation process.

The asymmetry of the CL band, as well as the peak at 0.98 eV can be considered the fingerprint of the presence of shallow levels induced by the ripplocations. This observation is in agreement with previous observations (for example, in the case of silicon, a material with a similar indirect bandgap), on the influence of strain fields on broadening, quenching and redshifts of the indirect bandgap transitions[21]. Similar effects are reported also for strained III–V direct bandgap semiconductors[22,23]. However, it must be stressed that in all the considered cases, the peak symmetry results unaltered by the presence of strain fields.

## Discussion

In conclusion, this work reports on the experimental evidence, confirmed by DFT calculations, of the unexpected observation of a novel NIR emission in multilayer $MoS_2$ flakes, peaked at 0.75 eV due to sulfur vacancies. In particular, we find an excess of $V_s$ at the flake edges. On the basis of an extensive structural characterization, we could also correlate the excess of $V_s$ with the formation of $MoS_2$ nano-rolls on the flake's edges. In addition we report on the observation of the onset of a novel emission peaked at 0.98 eV caused by the presence of ripplocations in the analysed flakes. Since both the 0.75 and the 0.98 eV NIR emissions have never been observed in these materials[21], our findings can open a new perspective for $MoS_2$ flakes with thickness larger than one monolayer. Indeed, one can consider to engineer the optical properties of the $MoS_2$ flakes during the synthesis or by post-growth treatments because they are mediated by the most energetically favoured native defect (namely, the sulfur vacancy), as well by the easy formation of extended defects like ripplocations.

Understanding the spatial variations in electronic structure due to the presence of defects, impurities, edges and so on, and their impact on the optical/electronic properties is important for future device applications. Therefore, since the chalcogen vacancy has the lowest formation energy for all the 2D TMDCs[13] we can conclude that our findings are timely and the overall impact is not negligible having a general validity for this class of materials.

## Methods

**Sample preparation and structural characterization.** Commercial molybdenite purchased from Graphene Supermarked has been exfoliated by standard tape method. A complete structural characterization of the mineral, as obtained by X-ray diffraction, is reported in Supplementary Figs 11–13 and Supplementary Note 3.

**CL spectroscopy.** CL spectroscopy was carried out with a commercial Gatan MonoCL2 system, fitted onto an S360 Cambridge SEM. The CL system is equipped with a Ge photodiode for the near infrared range 750–1,700 nm (1.6–0.7 eV). The infrared detectors are amplified by means of a lock-in amplifier. The CL spectra and maps were collected at room temperature with an accelerating voltage of 10 keV, a beam current of 10 nA and a spectral resolution of 5 nm (about 50 meV).

The single-peak parameters are, thus, evaluated by a deconvolution procedure using a standard Levenberg–Marquardt algorithm for the minimization of the $\chi^2$. To avoid any possible artefacts the fitting parameters peak position ($x_C$) and amplitude ($A$) were left free, while constraints were applied to the FWHM ($w$). We impose a $w$ maximum equal to 0.5 eV. At the end all the peak positions are affected by an error of 0.01 eV, which is less than the error due to the spectral

resolution of the measurement. The amplitude and the FWHM have a relative error of the 5%.

**TEM analyses.** A JEOL 2200FS FEG STEM/TEM working at 200 kV was used for the high-resolution TEM experiment. The atomic resolution HAADF images and the EELS spectra reported in this article were obtained at 60 kV using the JEOL ARM200F sub-Angstrom microscope installation at Beyond-Nano Lab. This consists of a probe-corrected microscope equipped with a C-FEG and a fully loaded GIF Quantum ER as EELS spectrometer.

**Raman spectroscopy and mapping.** Raman spectroscopy was performed by using a Renishaw Invia spectrometer equipped with the 2.4 eV/514 nm line of an $Ar^+$ laser line focused on the sample by a $\times 100$ and 0.90 numerical aperture Leica objective providing a spot diameter of about 0.7 µm. The power at the sample was maintained at 1 mW to prevent laser-induced sample heating and desorption. All the measurements were carried out in a z-backscattering geometry. The Raman map has been acquired on a $5 \times 5\,\mu m^2$ area with a step of 0.6 µm. Supplementary Fig. 14 reports the Raman maps of the $E_{2g}$ and $A_{1g}$ mode positioning used to identify the flake area in correspondence with the optical microscopy image in the Raman spectroscopy analysis. The $A_{1g}/E_{2g}$ intensity ratio has been obtained by fitting the acquired spectra in each point with pseudo-Voigt (product of Lorentzian and Gaussian functions) curves.

**DFT calculations.** The DFT calculations were performed with the PBE[17] exchange and correlation functional supplemented by a semiempirical van der Waals correction according to Grimme et al.[24]. We used norm-conserving pseudopotentials and a plane wave expansion of the Kohn–Sham orbitals with a 90 Ry cutoff by using the Quantum-espresso suite of programmes[25]. (www.quantum-espresso.org). Brillouin zone integration was performed over a $6 \times 6 \times 6$ or $4 \times 4 \times 6$ uniform mesh. The lattice parameters have been fully relaxed and change from $a = 3.193$ Å and $c = 12.450$ Å for the ideal system with no vacancies to $a = 3.185$ Å and $c = 12.426$ Å and $a = 3.180$ Å and $c = 12.409$ Å for sulfur vacancy concentrations of 2.1 atom% and 3.7 atom%, respectively.

**Data availability.** The data that support the findings of this study are available from the corresponding author on request.

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

## Acknowledgements

This work is partly supported by the U.S. Army Contract W911NF-14-1-0612, by the European Project SYNAPSE and by the Project BioNiMed. Part of this work has been performed at Beyondnano CNR-IMM, Catania, Italy, which is supported by the MIUR under project Beyond-Nano (PON a3_00363). A.M. and E.C. acknowledge Dr C. Martella (CNR-IMM) for his support in the analysis of the Raman maps.

## Author contributions

G.S. conceived the experiments, wrote and revised the paper and supervised the entire work; F.F. conceived and performed the CL experiments, wrote and revised the paper and elaborated the optical data; V.S. contributed to the experiments, to the paper drafting and revision, and supervised the rebuttal letter; D.K. contributed to Raman experiments and to the DFT calculations; A.M. supervised the Raman spectroscopy and imaging, and contributed to the paper drafting and revision and to the rebuttal letter; E.C. performed the Raman spectroscopy and imaging; M.L. contributed to the experiments and to the paper drafting; L.L. supervised the transmission electron microscopy experiments and the imaging elaboration, and contributed to the paper drafting; M.B. performed and supervised the DFT calculations and contributed to the rebuttal letter; E.R. performed and elaborated the transmission electron microscopy experiments; G.N. co-performed the STEM experiments in the sub-Angstrom microscope; C.F. and E.B. performed the X-ray experiments; D.C. performed DFT calculations.

## Additional information

**Competing financial interests:** The authors declare no competing financial interests.

