## [Peer review file · Nature Communications]

Reviewers' comments:

Reviewer #1 (Remarks to the Author):

The work reports a detailed study of the effect of defects and impurities on the optical properties of few-layer MoS₂ using several spatially-resolved techniques. The authors have put in extra effort to improve the work and many of the technical issues have been clarified in the revised manuscript. I suppose it would be suitable for publication in Nature Communications if the authors could address the following issues.

1) The novelty and potential impact of the work

I do not think the novelty and potential impact of the work lies in the discovery of the new NIR emission features. The emission quantum yield of these defect features is very low as quantified by the authors as only 30% more than the weak indirect band-to-band recombination. In addition, these emission features cannot be observed by photoexcitation. The authors speculated two possible reasons, the first being their low emission quantum yield and the second being electron-beam induced S vacancies as their origin. None of these make the defect features interesting.

2) The authors should provide the sample thickness.

3) The authors should address whether the reported phenomena have been observed in multiple samples and the role of electron beam exposure on the observed phenomena.

Reviewer #2

In the originally submitted manuscript, authors Fabbri, et. al. studied MoS₂ flakes with cathodoluminescence spectroscopy, and observed a 0.75eV peak due to sulphur vacancies. In the original review process, a reasonable number of objections were raised, which needed to be resolved prior to publication.

In my opinion, the authors have now addressed these objections to my satisfaction in the revised manuscript:

1. **Regarding the motivation/importance of their specific result:** They have now resolved this by adding appropriate sentences in the introduction about the importance of defect engineering in MoS₂ flakes and how their results validate such a possibility. I am reasonably satisfied.
2. **Regarding the absence of S versus an abundance of Mo:** The authors have added a paragraph to the text clarifying that it is difficult to absolutely distinguish between S deficiency versus Mo increase. However, they've provided satisfactory qualitative arguments related to chemistry and energy-minimization, which support S deficiency.
3. **Regarding the analysis of their quantitative data:** The authors have now provided discussions about error bars to distinguish between their different energy peaks. They have also clarified, simplified and re-explained some of the analysis related to the choice of regions in Fig2, which notes the observation of S deficiency.
4. **Regarding the analysis and interpretation of their 1.07eV peak:** The authors have clarified the reasons for not being able to spatially resolve the 1.07eV, as well as why the peak may not have been seen/reported in previous literature.

In conclusion, I am satisfied with the answers provided by the authors, and I support its publication. I continue to think that this direction of study – using spatially-resolved techniques to study the effect of defects and impurities on the electronic and optical properties, is an important direction for 2D van der Waals materials. I wish them well in the further continuation of their work.

Authors' response to Reviewer's comments:

Comment #1 on *the novelty and potential impact of the work*: I do not think the novelty and potential impact of the work lies in the discovery of the new NIR emission features. The emission quantum yield of these defect features is very low as quantified by the authors as only 30% more than the weak indirect band-to-band recombination. In addition, these emission features cannot be observed by photoexcitation. The authors speculated two possible reasons, the first being their low emission quantum yield and the second being electron-beam induced S vacancies as their origin. None of these make the defect features interesting.

We sincerely appreciate the reviewer's comments and his/her astute observations on our finding of the near-infrared emission peak ~ 0.75 eV attributed to Sulfur vacancy, V_S , in exfoliated MoS_2 flakes. We respectfully disagree with the reviewer's assessment of the lack of novelty and potential impact of our result for the reasons enumerated herein:

1) To the best of our knowledge, hitherto, there has been no report of a direct identification of a luminescence band with a native defect in any of the 2-dimensional transition metal dichalcogenides. Our paper for the first time makes an unambiguous identification of an infrared (IR) luminescence band with a native defect in MoS_2 , such as V_S .

2) It is indeed a rare phenomenon to observe radiative recombination activity of a deep level caused by native defects (vacancies, interstitials and antisites) in semiconductors. Generally these native defects act as non-radiative centers and lifetime killers. In this regard, to observe a luminescence band due to a native defect in MoS_2 flakes is indeed surprising and novel. Further, calculations of the charge transition level within density-functional theory with a hybrid exchange-correlation functional [in Ref. 15 of the manuscript] revealed that the single sulfur vacancy induces an acceptor state at about 0.77 eV below the conduction band for bulk MoS_2 . This energy can be taken as an estimate of the adiabatic transition energy from the conduction band to the defect state which, in the case of a small Franck-Condon shift, would be comparable to the vertical transition energy measured by CL, lending credence to the assignment of the 0.75 eV emission with V_S . [This point is clarified in the revised manuscript, p.8, lines 15-18.]

3) Yes, it is true that the 0.75 eV emission is only $\sim 30\%$ higher than the indirect band-to-band recombination. Nonetheless, we are of the opinion that our finding can open a new inquiry in the current research of MoS_2 photonics because we observe luminescence activity (stronger than the phonon mediated indirect band gap transition) from native defects that are generally expected to be nonradiative.

4) Yes, it is true that the 0.75 eV feature, or even the 0.98 eV emission attributed to ripplons, are observed by electron excitation of the luminescence. This does not, however, preclude their observation by photoexcitation. It only shows that stable configurations of native defects such as V_S have to be introduced in MoS_2 flakes by equilibrium methods (e.g. thermal annealing in controlled Sulfur partial pressures). In fact, thermal annealing of monolayer of MoS_2 affects profoundly its luminescence (Ref: "Excitation intensity dependent photoluminescence of annealed two-dimensional MoS_2 grown by chemical vapor deposition," D. Kaplan, K. Mills, J. Lee, S. Torrel, and V. Swaminathan, *Journal of Applied Physics* 119, 214301 (2016) and other references therein).

5) The novelty of the 0.75 eV emission, attributed to V_S , albeit its current low efficiency should not deter one to explore further the exciting opportunity of creating V_S rich MoS_2 flakes in order to enhance the radiative efficiency. In this regard, a recent result on Electron-Beam Induced

Transformations of Layered Tin Dichalcogenides (E. Sutter, Y. Huang, H.-P. Komsa, M. Ghorbani-Asl, A.V. Krasheninnikov and P. Sutter, accepted in Nano Lett (communicated to us by author AVK)) demonstrates that by controlled removal of chalcogen atoms in SnS₂ and SnSe₂ can give rise to ordered nonmetal vacancy lines and convert the parent material to monochalcogenides. This prompts the exciting possibility of creating nonmetal vacancy lines (or even a vacancy sublattice) in MoS₂, thereby giving rise to collective behavior and increased radiative efficiency. Our result opens up such explorations of 'vacancy engineering' of transition metal dichalcogenides for exciting new fundamental science as well as for technological applications as suggested below.

6) The 0.75 eV (1650 nm) emission falls in the Short Wave IR band which is a technologically important spectral range for a range of applications such as light emitters and detectors. In particular, for the detector applications, the currently used InGaAs based systems are prohibitively expensive for large array formats. Our result can potentially pave the way for engineering a low cost SWIR detector using 'vacancy engineered' MoS₂ or other transition metal dichalcogenides or even monochalcogenides.

7) In closing, we have compellingly showed that a close tuning of the S deficiency in MoS₂ can be exploited to promote radiative processes in SWIR spectral range that has hitherto been unexplored. Further, we strongly believe that our finding may prompt new research/technology direction in the defect manipulation and engineering of transition metal dichalcogenides or even monochalcogenides.

Comment #2: The authors should provide the sample thickness

We thank the reviewer for this suggestion. The thickness calculation is now reported in the supplementary material in Fig. S3.1 where a STEM image of the edge of a flake is shown together with the intensity line profile. As for the edges of all the studied flakes, they are not abrupt but the thickness is slowly decreasing in a stepped fashion. We exploited this feature in order to evaluate the sample thickness by analyzing the different gray levels in the STEM micrographs, corresponding to different number of monolayers. The maximum thickness measured for the studied flakes, i.e. in the specimen center, does not exceed 30 ML (18 nm) and typically ranges between 30 and a few monolayers going toward the flakes' edges. However, it can be clearly observed in Fig. S3.1 that the lateral size of the thickness decreasing region is about 200 nm. This dimension is comparable with the region where the 0.75 eV emission is recorded. Therefore, we can conclude that the IR emission is a general feature of multilayer flakes, independent of the actual flake thickness.

We have added the above clarifications on the sample thickness in the body of the revised manuscript on page 6 at line 9. Specifically, the following sentence (highlighted in yellow in the revised manuscript) is added:

"We evaluated the samples thickness by analyzing the different gray levels in STEM micrographs, corresponding to different number of monolayers. The maximum thickness measured for the studied flakes, i.e. in the specimen center, does not exceed 30 ML (18 nm) and typically ranges between 30 and a few monolayers going toward the flakes borders. However, from Fig. S3.1 in Supplementary Information, it can be clearly observed that the lateral size of the thickness decreasing region is about 200 nm. This dimension is comparable with the region where the 0.75 eV emission is recorded."

Therefore, we can conclude that the IR emission is a general feature of multilayer flakes, independent of the actual flake thickness.”

Comment #3: The authors should address whether the reported phenomena have been observed in multiple samples and the role of electron beam exposure on the observed phenomena.

We thank the reviewer for raising the questions on the effect of beam exposure on the observed phenomena and the repeatability of the results.

A. We have clarified the effect of beam exposure in the Supplementary Information by including Fig. S2.4 and in the discussion pertaining to it. Our response is reproduced below:

“The effect of the electron beam exposure has been more carefully considered and studied by CL spectroscopy. In order to understand the effect of the electron beam irradiation on the stability of the different CL emissions reported in our manuscript, we irradiated a MoS₂ flake for 30 minutes in the SEM with the same parameters used for the CL characterization and then we acquired an additional CL spectrum. The intensity of both the 0.76 eV and 1.12 eV emissions decrease following 30 min irradiation. In addition, the electron beam irradiation mainly affects the high energy tail of the 1.12 eV emission, due to the indirect band-to-band radiative transition of MoS₂. This effect is consistent with the formation of nonradiative centers due to the electron beam irradiation, as recently reported in E. Rotunno et al., 2D Materials, 3(2), 025024 (2016), that affect the different emissions composing the CL spectrum.”

We have also added the sentence below in the revised manuscript, page 5, line 4, to draw the reader’s attention to the Supplementary Information.

“In addition, the stability of the light emissions under electron beam irradiation is tested (See Supplementary Information, Fig. S2.4) by means of CL spectroscopy after a 30 minutes of irradiation of a MoS₂ flake.”

Figure S2.4 CL spectra of MoS₂ flake before (red line) and after (green line) 30 minutes electron beam irradiation.

B. Concerning the reviewer's comment about the observation of the 0.75 eV emission in different samples, the emission appears in all the twenty flakes investigated as well as in the cracked molybdenite (see Supplementary Information, Fig. S1.5) analyzed by CL spectroscopy and imaging. While the 0.75 eV emission is observed in all the samples examined, variations of its intensity were noted among different flakes. For example, Figure S2.3 of the Supporting Information reports the results from a flake with an interesting distribution of the 0.76 eV emission that differs from the one in the manuscript. We have given below an additional example showing the CL mapping acquired at about 1.10 eV and at 0.76 eV supporting the reliability of the results.

a) SE image

b) 1.10 eV monochromatic CL map

c) 0.76 eV monochromatic CL map

Figure a) SE image of the MoS₂ flake under analysis, in the low right side of the image the TEM copper grid appears, b) and c) 1.07 eV and 0.76 eV monochromatic CL maps respectively.